# GeoMath🖼️: A Benchmark for Multimodal Mathematical Reasoning in Remote Sensing

## Abstract

Vision-language models (VLMs) have demonstrated impressive performance in various Earth observation tasks, particularly in zero-shot capabilities. However, their mathematical reasoning skills in remote sensing (RS) remain unexplored due to the lack of relevant data. To close this gap, we introduce GeoMath, a multimodal mathematical reasoning benchmark meticulously designed for the RS domain. It comprises 3773 high-quality vehicle-related questions from aerial perspectives, spanning 6 mathematical subjects and 20 topics. All data used in this benchmark were collected by our drones from various altitudes and perspectives. Despite the limited geographical coverage, full access to all parameters of the RS images and detailed vehicle information ensures that the constructed mathematical problems are rigorous and diverse. With GeoMath, we have conducted a comprehensive and quantitative evaluation of 14 prominent VLMs. Solving these math problems requires high-resolution visual perception and domain-specific mathematical knowledge, which poses a challenge even for state-of-the-art VLMs. We further explore the impact of image resolution and the zero-shot prompting strategy on the scores, analyzing the reasons behind GPT-4o's reasoning errors. By comparing the gap between InternVL2 and GPT-4o, we find that the latter exhibits some level of cross-view knowledge transfer capability.

## 1 Introduction

Deep learning has achieved significant success in remote sensing (RS), but it often faces safety concerns due to its black-box nature (Höhl et al., 2024). The advent of vision-language models (VLM) (Yin et al., 2023), which exhibit strong mathematical reasoning capabilities, offers a new approach to developing reliable RS interpretation systems (Wang et al., 2024c). VLMs can emulate human-like visual reasoning by employing a visual encoder to act as the "eyes" for perception and leveraging a large language model (LLM) as the "brain" for analysis (Dasgupta et al., 2022), facilitating seamless information transfer between visual and textual modalities. Unlike traditional deep learning models, VLMs can offer a transparent reasoning process. To ensure the development of trustworthy RS interpretation systems, it is crucial to rigorously assess the multimodal mathematical reasoning abilities of VLMs.

Numerous RS Visual Question Answering (VQA) datasets (Lobry et al., 2020; Zheng et al., 2021; Zhang et al., 2023a) have been created to evaluate the capabilities of multimodal question answering systems. However, most of these questions primarily assess the model's visual perception abilities, with math-related questions representing only a small fraction. These math questions are often limited to counting and 2D spatial relationships, leaving the model's broader mathematical reasoning capabilities largely unexplored. Moreover, since these questions can be answered without domain-specific knowledge (e.g. metric geometry, imaging principles, perspective transformation), they inevitably lack specialization. Hence, there is a pressing need to (1) establish a new benchmark that requires domain-specific knowledge, to facilitate the development of RS VQA systems, and (2) assess the progress of vision-language geofoundation models (VLGFMs) (Zhou et al., 2024), especially their mathematical reasoning capabilities.

In this paper, we present GeoMath, a multi-modal mathematical reasoning benchmark within the context of remote sensing imagery. It encompasses six mathematical subjects: *geometry*, *logic*, *statistics*, *arithmetic*, *counting*, and *algebra*. The benchmark supports five potential application

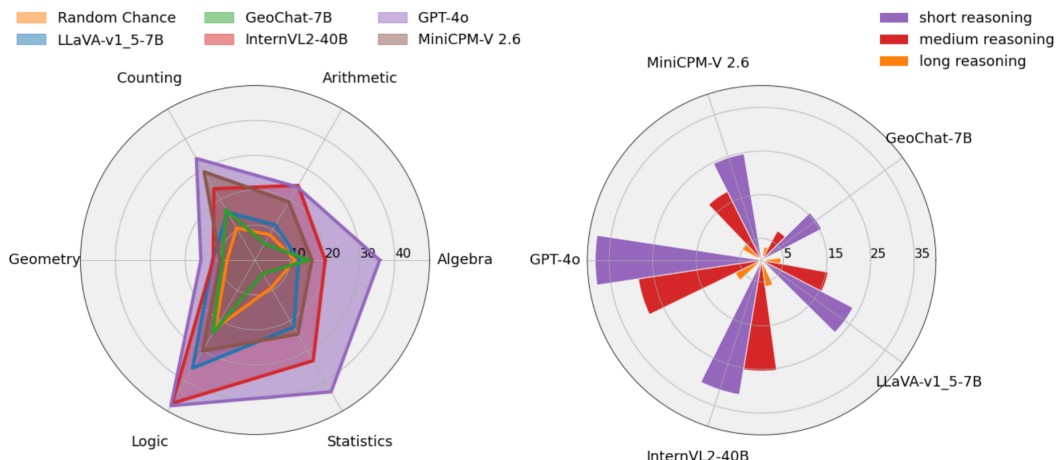

Figure 1: Accuracies of four leading VLMs, one VLGFM, and random chance on our proposed GEOMATH across mathematical subjects and reasoning steps.

scenarios: *surveying*, *surveillance*, *market research*, *entertainment*, and *military*. Each question in the benchmark provides detailed reasoning steps, with the minimum reasoning step size being 2 and the maximum being 6. The benchmark covers 11 distinct 4K resolution RS scenes, with varying combinations of drone's above ground level (AGL) and pitch angles. In general, GEOMATH comprises 3,773 newly created problems (Table 1). For fine-grained evaluation, the examples are annotated with metadata, including question type, answer type, rationale, reasoning steps, pitch angle, AGL, and necessary context. A detailed description of data collection can be found in §2.

We conducted extensive experiments in GEOMATH to evaluate the reasoning abilities of 14 foundation models, which exhibit state-of-the-art performance in multimodal reasoning tasks. Among these models, GPT-4o (OpenAI, 2023) is a proprietary model and GeoChat (Kuckreja et al., 2024b), is fine-tuned in RS data. Furthermore, we explore several zero-shot prompting techniques to shift the model from a single-step reasoning paradigm to a multi-step reasoning mode, aligning more closely with human cognitive processes. It includes Chain-of-Thought (CoT) (Wei et al., 2022) and Plan-and-Solve (PS) (Wang et al., 2023) designed for LLMs, as well as Description CoT (DespCoT) (Wu et al., 2023) and Compositional CoT (CCoT) (Mitra et al., 2024) tailored for VLMs.

To our knowledge, we have taken a meaningful first step towards multimodal mathematical reasoning in RS. This work selects vehicles as the main subject and provides a preliminary exploration of mathematical problems in remote sensing, without involving multisource RS images or complex sensor characteristics. As illustrated in Figure 1, GPT-4o demonstrates superior performance in five subjects. However, even the highest overall accuracy achieved is only 34.6%. We highlight the challenges that high-resolution RS images pose to VLMs. Our in-depth analysis in §3.3 and E.6 reveals that the knowledge transfer capabilities of GPT-4o are another key factor contributing to its superior performance in GEOMATH. We hope that GEOMATH will serve as a valuable resource, providing a benchmark for the future development of trustworthy multimodal interpretation systems of RS.

## 2 THE GEOMATH DATASET

As mentioned above, there is a noticeable gap in the RS VQA benchmarks, which mainly focus on evaluating the perceptual capabilities of models while neglecting the mathematical capabilities. Therefore, our dataset, GEOMATH, aims to bridge this gap by providing a robust evaluation benchmark for mathematical reasoning intertwined with RS visual perception. In this section, we present the GEOMATH, following the steps of data collection, metadata annotation, question design, and question generation. Finally, we perform data analysis on the dataset.

### 2.1 DATA COLLECTION

To the best of our knowledge, there is currently no dedicated mathematical dataset specifically designed for remote sensing. Existing open-source RS datasets (Xia et al., 2018; Li et al., 2020) often lack sensor metadata and provide limited target attributes. Consequently, these datasets can only

support the formulation of simple mathematical problems, such as counting the object according to its color or judging the relative position in the image. To develop a more specialized and diverse mathematical dataset, we use unmanned aerial vehicles (UAVs) to collect data from scratch. This approach ensures comprehensive access to sensor parameters and detailed information about ground targets. To enhance the diversity of mathematical problems, we choose vehicles as the subject of drone photography. Compared to buildings or land cover (Yang & Newsam, 2010), vehicles have richer attributes and more fine-grained categories. Data collection is divided into two parts: aerial imagery and ground video.

**Aerial Imagery.** All aerial images in GEOMATH were collected with a small UAV platform, DJI Mini3, between 10-16 September 2023, in Shanghai. The dataset consists of 4K high-resolution RS images from 11 distinct scenes, captured at 9 different above-ground levels (AGLs) and 3 pitch angles. This implies that these RS images possess different spatial resolutions and perspectives. In addition, the collected images cover a variety of weather scenarios, such as sunny, cloudy, and rainy days, along with different lighting conditions. Details are provided in §B.1.

**Ground Video.** We record ground videos from the same areas to facilitate accurate annotation of vehicle brands and models. Specifically, we select time slots with relatively low vehicular mobility, avoiding rush hours and meal times. Additionally, to mitigate the vehicle mismatch between drone images and ground videos caused by vehicle entry and exit, we capture two sets of ground videos before and after the drone captures aerial photos. This ensures that vehicles entering or exiting the scene halfway through the capture are recorded in the videos. However, there are instances where vehicles pass through the scene briefly, leading to cases where they are not captured in either video. In such situations, we mask these vehicles with a black mask in the images to ensure that all visible vehicles have fully known attributes. Due to privacy concerns, ground videos will not be released.

## 2.2 METADATA ANNOTATION

The metadata we use can be categorized into two main components. The first includes camera-related parameters, such as intrinsic parameters (focal length, pixel size, sensor dimensions) and extrinsic parameters (pitch angle, AGL). These are extracted from the raw data from the drone. The second component pertains to vehicle fine-grained attributes, which require manual annotation. To accurately describe the length and width of vehicles, we use rotated bounding boxes to annotate their positions (Yang et al., 2022). Then, a 360 degree angle representation is used to depict the vehicle's orientation (Hu & Tong, 2023). Identifying specific vehicle brands from aerial imagery presents a significant challenge for human annotators, and as a result, existing publicly available RS vehicle datasets have not achieved brand-level annotations (Mundhenk et al., 2016; Zhu et al., 2021).

However, leveraging the previously mentioned ground videos, we successfully created the first RS vehicle dataset with fine-grained attributes, identifying vehicles down to the model level within each brand. Specifically, we match the vehicles in the aerial image with the vehicles in the ground video one by one according to their locations, and then call the DCD's API [1] to identify the specific model based on the vehicle's appearance and logo in the ground image. For vehicles whose models could not be identified, we used a black mask to cover them from the image. Then we used the DCD car database to obtain detailed attributes, such as the size and price of each car. Vehicle prices were sourced during August 2024, and the average price is calculated based on the maximum and minimum market values. With detailed vehicle attributes and sensor parameters (§B.2), GEOMATH can be established. In the next subsection, we will list the metadata used for each type of question.

## 2.3 QUESTION DESIGN AND GENERATION

Recent works (Li et al., 2024; Xu et al., 2024) adpot GPT to automatically generate RS VQAs, to reduce manual labor. Compared to template-based methods, model-generated questions exhibit greater diversity. However, in mathematical benchmarks, the rigor of the questions is paramount. Given the current performance of GPT on multimodal mathematical benchmarks (Lu et al., 2024b; Wang et al., 2024b), we cannot fully trust it. Therefore, we choose a template-based question gen-

---

[1] https://dcdapp.com

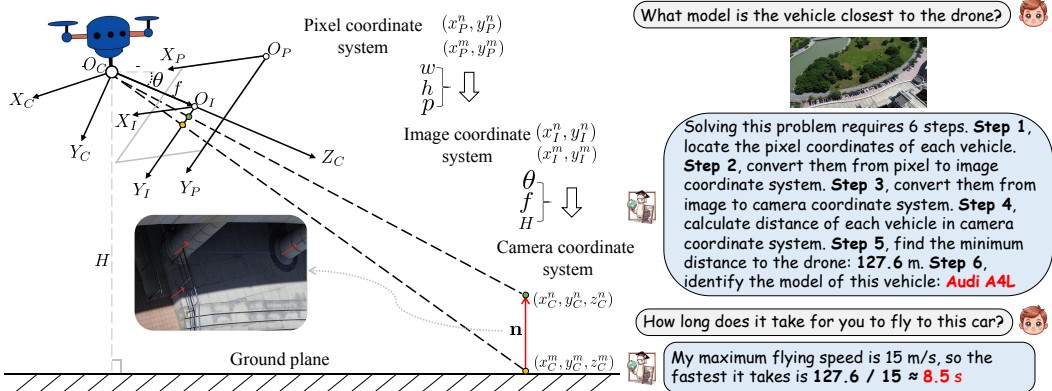

Figure 2: Mathematical modeling of UAV Scenes and examples for geometric question.

eration approach, which offers more control over content compared to generative models. To compensate for the lack of diversity, we design more than 80 templates based on 20 topics(A.3).

**Geometry.** The geometric questions extend the spatial relationships in RS VQA from the 2D pixel plane to the 3D real world. The related camera parameters include pitch angle $\theta$, AGL $H$, focal length $f$, and pixel size $p$. Relevant domain knowledge includes metric geometry and prospective geometry. Figure 2 illustrates a typical UAV reconnaissance scenario. Given the relatively flat terrain of the shooting area, we can assume that it satisfies the assumption of a flat surface (Novak, 2017). We validated this assumption by placing normal vectors **n** on reference objects such as poles. Given the camera parameters, the pixel coordinates of a car can be used to compute its camera coordinates. The complete calculation formulas are detailed in §B.3. Based on these, the closest vehicle can be identified and the shortest flight time can be estimated based on the speed of the drone. In addition, we can estimate the area of the captured region as well as the size and orientation of the vehicles.

**Arithmetic.** We construct a series of arithmetic questions, including addition, subtraction, multiplication, and division, based on the prices of the vehicles. For example, questions may ask which of the two cars is more expensive or how many of a certain type of car can be bought with 1 million RMB. Considering that vehicle prices can be unstable due to market fluctuations, we have provided the vehicle models and their corresponding prices in the context field of each problem. We exclude questions that can be answered solely through pure text, ensuring that the model must rely on visual data to arrive at the correct answer. This approach ensures that the model can obtain all the necessary information to solve the current mathematical problems in an offline environment, without the need for retrieval-augmented generation (RAG) (Gao et al., 2023) techniques.

**Counting.** By incorporating more fine-grained attributes of the cars, we are able to construct a wider variety of counting questions with varying levels of difficulty. Related attributes includes vehicle types, brands, models, and prices. The generated questions not only involve counting based on single-attribute constraints but also include comparative counting and counting based on multiple attribute constraints. For example, questions may ask for the number of cars priced above 100,000 RMB or the number of white SUVs. In GEOMATH, each image contains an average of 25.8 cars. The differences between vehicles are smaller compared to those between different object categories, making the task more challenging.

**Algebra.** The algebraic questions are primarily divided into two categories: single-variable algebra and multi-variable algebra. The model needs to use its visual perception capabilities to obtain certain variables and then solve equations to determine the target variable. The relevant domain knowledge includes spatial coordinate system transformations, such as determining the coordinates of a vehicle in the image or camera coordinate system based on its pixel coordinates obtained from the image. We also construct algebraic questions related to prices, such as calculating the price of the vehicle closer to the image bottom based on the total price of two cars and their price ratio.

**Logic.** In the design of logic problems, beyond incorporating image-based information, some common-sense knowledge from daily life is introduced. For example, electric vehicles do not need to visit gas stations regularly, and the number of passengers a taxi can accommodate is equal to the total number of seats in the vehicle minus one (excluding the driver).

| Statistic | Number |
|---|---|
| Total questions | 3,773 |
| - Multiple-choice questions | 1,352 (35.8%) |
| - Free-form questions | 2,181 (57.8%) |
| - True/False questions | 240 (6.4%) |
| Unique number of images | 360 |
| - Pitch angle: 90 | 117 (32.5%) |
| - Pitch angle: 60 | 126 (35%) |
| - Pitch angle: 45 | 117 (32.5%) |
| - Above ground level: low | 138 (38.3%) |
| - Above ground level: medium | 108 (30.0%) |
| - Above ground level: high | 114 (31.7%) |
| Unique number of questions | 424 |
| Unique number of answers | 686 |
| Maximum question length | 236 |
| Minimum question length | 45 |
| Average question length | 101.5 |
| Maximum reasoning steps | 6 |
| Minimum reasoning steps | 2 |
| Average reasoning steps | 3.34 |

Table 1: Key statistics of GEOMATH.

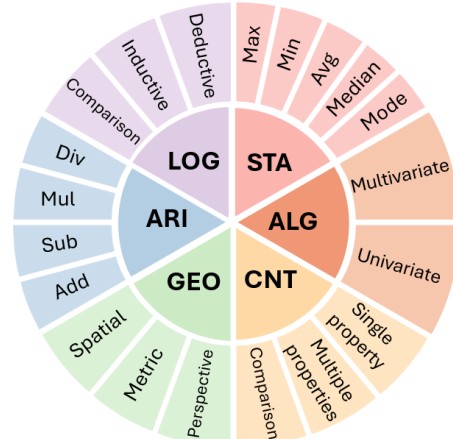

Figure 3: Question types covered by GEOMATH. There are 6 subjects and 20 topics in our benchmark. ARI: arithmetic, CNT: counting, ALG: algebra, STA: statistics, LOG: logic, GEO: geometry.

**Statistics.** We design statistical questions based on vehicle prices and sizes, covering maximum, minimum, mean, and mode. Related domain knowledge is metric geometry.

Existing RS VQA tasks focus mainly on single-step reasoning (Lobry et al., 2020), such as land cover and building classification. Our benchmark emphasizes multistep reasoning ability (Chen et al., 2024), with the minimum reasoning steps for all questions being 2 and the maximum being 6. As shown in Figure 1, the longer reasoning steps place higher requirements on the model's reasoning capabilities. We are the first RS VQA dataset that provides multistep reasoning processes for each question. Although it offers a feasible solution approach, it is not necessarily the only one. For the sake of rigor, the reasoning steps are not used to calculate the model scores. However, they can serve as a reference to help in analyzing the reasons behind the model's reasoning errors (see §3.4).

During the question generation phase, we prioritize the selection of images in which vehicles are not significantly occluded by buildings or trees to build the benchmark. The process consists of three steps: 1) generating image-level questions without modifying the images; 2) generating single-instance questions by randomly selecting a vehicle and drawing a rotated bounding box around it as a visual prompt; and 3) generating two-instances questions by randomly selecting two vehicles and drawing their rotated bounding boxes in different colors (e.g. red and blue). Vehicles near the edge of the image are excluded to avoid difficulties due to incomplete visual information. Finally, the generated questions are manually reviewed for accuracy.

## 2.4 DATA ANALYSIS

The main statistics of GEOMATH are presented in Table 1. There are three types of questions: multiple choice, free-form, and Ture or False. The answers to free-form questions are classified as integers, floating numbers, lists, or strings. Variations in pitch angle and AGL ensure the diversity of observation patterns in GEOMATH. The examples in §A.2 illustrate the various types of math problem. The comparison of the reasoning steps in Figure 9 with other RS VQA datasets highlights the complexity of the problems GEOMATH. More details on data analysis are available in §C.

## 3 EXPERIMENTS

GeoChat (Kuckreja et al., 2024a) has shown that fine-tuning VLMs on RS datasets enhances their generalization capabilities across various multimodal RS tasks. Our objective is to perform qualitative and quantitative analyzes using GEOMATH to assess whether this generalization extends to multimodal RS tasks that require specialized knowledge. §3.1 outlines our evaluation strategy, while

| Model | LLM | ALL | Subject | | | | | | AGL | | | Pitch Angle | | | Type | | |
|---|---|---|---|---|---|---|---|---|---|---|---|---|---|---|---|---|---|
| | | | ALG | ARI | CNT | GEO | LOG | STA | Low | Med | High | 45 | 60 | 90 | FRE | CHO | T/F |
| Random chance | - | 11.7 | 11.5 | 8.4 | 10.6 | 8.0 | 22.6 | 9.2 | 10.7 | 13.2 | 12.3 | 11.1 | 12.5 | 12.2 | 0.0 | 24.3 | 51.3 |
| *Small-scale VLMs (LLM's Parameters < 10 Billion)* | | | | | | | | | | | | | | | | | |
| GeoChat | Vicuna-7B | 12.6 | 15.4 | 5.4 | 16.7 | 9.4 | 24.1 | 4.5 | 13.9 | 12.7 | 10.7 | 11.5 | 12.3 | 13.7 | 4.6 | 19.9 | 42.1 |
| XComposer2 | InternLM2-7B | 13.6 | 4.0 | 2.9 | 20.0 | 2.3 | 28.7 | 23.4 | 15.4 | 13.5 | 12.0 | 12.0 | 14.4 | 14.7 | 11.0 | 11.8 | 49.6 |
| Qwen-VL-Chat | Qwen-7B | 16.5 | 6.7 | 10.7 | 15.8 | 11.4 | 30.2 | 24.1 | 18.8 | 18.1 | 17.8 | 18.1 | 18.4 | 18.3 | 9.5 | 26.9 | 49.2 |
| LLaVA-v1.5-7B | Vicuna-7B | 18.3 | 12.5 | 11.6 | 15.8 | 12.2 | 35.8 | 22.2 | 19.1 | 22.7 | 18.3 | 19.9 | 18.5 | 21.4 | 10.9 | 28.3 | 54.2 |
| XComposer2.5 | InternLM2-7B | 18.5 | 4.0 | 16.4 | 26.1 | 7.2 | 26.8 | 30.3 | 20.4 | 18.6 | 17.6 | 18.4 | 20.0 | 18.5 | 7.7 | 30.8 | 55.0 |
| DeepSeek-VL | DeepSeek-7B-Base | 18.5 | 9.2 | 18.5 | 18.9 | 8.8 | 28.7 | 27.2 | 22.1 | 18.6 | 17.4 | 19.4 | 20.2 | 19.0 | 8.8 | 30.9 | 53.8 |
| MiniCPM-V 2.5 | Llama3-8B | 20.0 | 21.5 | 11.8 | 24.2 | 9.2 | 29.9 | 23.2 | 20.4 | 20.1 | 17.7 | 19.0 | 19.6 | 19.8 | 6.5 | 33.3 | 59.6 |
| MiniCPM-V 2.6 | Qwen2-7B | 21.6 | 16.3 | 19.3 | 29.2 | 10.0 | 30.1 | 24.5 | 24.3 | 18.6 | 20.1 | 19.2 | 23.3 | 21.0 | 9.6 | 34.5 | 51.3 |
| InternVL2-8B | InternLM2.5-7B-Chat | 23.7 | 7.3 | 22.6 | 24.4 | 13.9 | 34.8 | 39.1 | 27.2 | 25.0 | 22.0 | 23.7 | 24.1 | 26.9 | 12.0 | 38.2 | 66.3 |
| *Large-scale VLMs (LLM's Parameters > 10 Billion)* | | | | | | | | | | | | | | | | | |
| LLaVA-v1.5-13B | Vicuna-13B | 17.2 | 10.2 | 18.7 | 19.2 | 11.8 | 22.8 | 20.3 | 17.2 | 18.0 | 19.5 | 18.3 | 18.4 | 17.8 | 5.7 | 33.1 | 47.5 |
| InternVL-Chat-V1.5 | InternLM2-Chat-20B | 18.8 | 17.1 | 13.4 | 18.9 | 9.6 | 30.4 | 23.4 | 20.7 | 17.2 | 19.0 | 16.3 | 19.7 | 21.3 | 8.1 | 29.7 | 59.2 |
| LLaVA-v1.6-34B | Hermes-Yi-34B | 23.9 | 12.1 | 17.7 | 31.7 | 15.1 | 37.0 | 29.6 | 26.1 | 24.6 | 21.6 | 22.9 | 25.6 | 23.9 | 10.6 | 39.4 | 61.7 |
| InternVL2-40B | Nous-Hermes-2-Yi-34B | 26.8 | 20.1 | 24.7 | 23.6 | 12.0 | 47.4 | 33.3 | 30.1 | 27.4 | 24.5 | 25.3 | 29.8 | 27.2 | 15.9 | 40.5 | 59.6 |
| GPT-4o | - | 33.5 | 35.7 | 24.2 | 33.6 | 15.5 | 48.2 | 43.6 | 36.8 | 31.7 | 29.9 | 30.7 | 34.1 | 34.2 | 18.8 | 50.8 | 62.5 |
| *Zero-Shot Prompting Technique* | | | | | | | | | | | | | | | | | |
| CoT (LLaVA-v1.6-34B) | Hermes-Yi-34 | 20.7 | 14.2 | 14.6 | 28.3 | 10.9 | 34.1 | 22.1 | 22.0 | 18.9 | 20.9 | 19.2 | 20.5 | 22.4 | 9.3 | 34.0 | 49.2 |
| CoT (InternVL2-40B) | Nous-Hermes-2-Yi-34B | 30.2 | 22.8 | 25.5 | 35.6 | 11.1 | 49.4 | 36.8 | 32.0 | 28.5 | 28.2 | 29.0 | 31.4 | 28.6 | 16.6 | 44.2 | 68.3 |
| CoT (GPT-4o) | - | 34.1 | 32.8 | 23.9 | 34.4 | 14.9 | 51.3 | 47.1 | 36.6 | 33.0 | 31.5 | 32.0 | 33.5 | 36.3 | 20.7 | 49.0 | 69.2 |
| PS (InternVL2-40B) | Nous-Hermes-2-Yi-34B | 28.4 | 21.7 | 22.6 | 29.2 | 12.6 | 48.1 | 36.2 | 31.7 | 27.7 | 26.0 | 28.5 | 30.8 | 26.5 | 16.1 | 42.9 | 62.1 |
| PS (GPT-4o) | - | 34.6 | 35.3 | 24.2 | 32.5 | 14.5 | 55.1 | 45.8 | 38.4 | 31.9 | 32.0 | 33.3 | 35.1 | 34.8 | 20.5 | 50.7 | 68.8 |
| CCoT (InternVL2-40B) | Nous-Hermes-2-Yi-34B | 24.8 | 19.8 | 19.5 | 20.8 | 12.5 | 44.2 | 32.0 | 26.5 | 24.9 | 24.9 | 24.7 | 25.6 | 26.2 | 13.8 | 39.6 | 52.9 |
| DCoT (InternVL2-40B) | Nous-Hermes-2-Yi-34B | 25.0 | 20.1 | 19.2 | 23.3 | 12.5 | 41.9 | 33.0 | 27.3 | 25.5 | 24.0 | 25.5 | 25.0 | 26.7 | 14.9 | 37.1 | 60.4 |

Table 2: Accuracy scores on the GEOMATH. ALL: average accuracy of the six subjects. Mathematical subjects: ALG: algebra, ARI: arithmetic, CNT: counting, GEO: geometry, LOG: logic, STA: statistics. Reasoning steps indicate the logical sequence of thoughts taken to solve this question. FRE: free-form question, CHO: multiple choice question, T/F: true or false question. The highest scores among models in each section and overall are highlighted in blue and red, respectively.

Section §3.2 details the VLMs evaluated. Quantitative results are presented in Sections §3.3, followed by a qualitative analysis result in §3.4.

## 3.1 EVALUATION PROTOCOLS

In the realm of multimodal mathematical reasoning benchmarks, such as MathVista (Lu et al., 2024b), GPT is used to derive answers from the responses of various models. However, frequent OpenAI API calls for each evaluation can incur substantial costs, challenging independent researchers. Another reason for not using GPT to extract answers is that most RS interpretation systems are typically deployed in offline environments. To reduce the barrier, we design a two-stage answer generation-extraction strategy. In the first stage, the model freely generates answers, focusing solely on reasoning without format constraints. In the second stage, the model extracts content in the specified format from its response, improving the accuracy of the format. This decoupling of reasoning and formatting allows us to extract the final answer in an offline environment using regular expressions. During question generation, the type of data for each answer is stored in the "eva" field. In the extraction phase, regular expressions are applied based on the answer type to retrieve the answer from the model's response. GEOMATH includes multiple-choice, free-form, and true/false questions, with free-form being strings, integers, floats, or lists. So, we use the accuracy scores as a metric for evaluation. This allows users to efficiently assess their model performance in GEOMATH locally using the evaluation function we provided. For details on the evaluation prompts and parameters, refer to §D.

## 3.2 EXPERIMENTAL SETUP

We evaluated the models in GEOMATH under three setups: (a) *Vision-language Foundation Models* that include general models such as LLaVA (Liu et al., 2023), Qwen-VL-Chat (Bai et al., 2023), XComposer2 (Zhang et al., 2023b), DeepSeek-VL (Lu et al., 2024a), InternVL (Chen et al., 2023), MimiCPM-V (Hu et al., 2024), GPT-4o (OpenAI, 2023) and remote sensing VLM GeoChat (Kuckreja et al., 2024b). (b) Zero-shot prompting setting with CoT (Wei et al., 2022), PS (Wang et al., 2023), DCoT (Wu et al., 2023) and CCoT (Mitra et al., 2024).

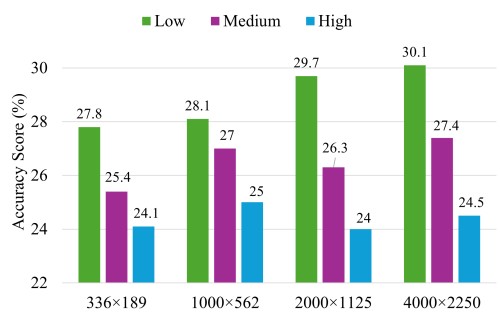
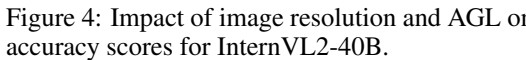

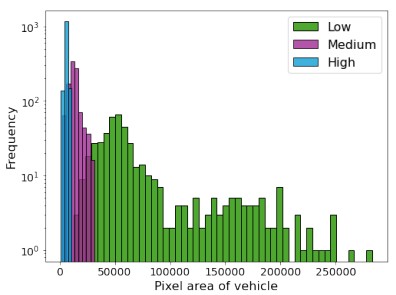

Figure 4: Impact of image resolution and AGL on accuracy scores for InternVL2-40B.

Figure 5: Distribution of pixel area occupied by vehicle under different AGLs.

## 3.3 EXPERIMENTAL RESULTS

Table 2 provides the performance results of various models, including prominent open source VLMs and the leading proprietary model. In light of literature Chen et al. (2024) indicating that LLMs exceeding 10 billion parameters emerge thinking and reasoning capabilities, we have categorized these models into two groups based on the size of their embedded LLMs to facilitate comparison. We create a random chance to serve as a reference baseline. A random option is selected for multiple choice and true/false questions, while free-form questions are left blank. We generate the random chance three times and average the results, then record in Table 2.

Among the VLMs evaluated, all models outperform random chance. Notably, InternVL2-8B achieved the highest score of 23.7 within small-scale models. Among the models that do not use zero-shot prompting, GPT-4o consistently achieves the highest overall score of 33.5. Although it falls behind InternVL2-40B in the arithmetic category, it retains a leading position in all other dimensions. Surprisingly, GeoChat (Kuckreja et al., 2024b), fine-tuned using LLaVA-v1.5-7B on RS data, exhibited a performance decline (for more details, see §E.4). To gain deeper insights into the reasoning capabilities of the model, we categorize the reasoning steps in Figure 1 into three groups: short (2 steps), medium (3-4 steps) and long (5-6 steps). The results indicate that the accuracy decreases sharply as the length of the reasoning steps increases.

For multiple-choice and true/false questions, models often do not require a full understanding of the domain-specific knowledge being tested. Instead, they can rely on logical reasoning and mathematical intuition to arrive at the correct answer. This approach may lead to a superficial understanding, where the model knows the correct answer without truly understanding the underlying concepts. To more accurately assess how well the models grasp RS expertise, we incorporated 57.8% free-form questions into GEOMATH, as shown in Table 1. These questions require the model first to extract the correct visual cues from the images and then to apply professional knowledge in remote sensing to calculate the precise answer, which makes them considerably more challenging. Among the models without using zero-shot prompting, GPT-4o achieves a free-form question score of 18.8, demonstrating the superior capability of GPT-4o.

**Impact of Image Resolution.** RS images of GEOMATH have a high native resolution of 4000×2250 pixels. In the previous experiments, the original 4K images were directly fed into the model without pre-processing. However, when these images are resized to the default resolution used by the models, such as 336×336 in LLaVA-v1.5, locating and counting vehicles becomes more challenging. To quantitatively examine the impact of image resolution on model performance, we performed a comparison experiment using the InternVL-40B model, which supports dynamic resolution technology (Liu et al., 2024a). Figure 4 illustrates how image resolution and AGL affect accuracy scores. We group AGL into three classes in ascending order: low (20-40m), medium (60-80m), and high (100-120m). Surprisingly, the big increase in resolution has little impact on the score. The results show that at lower altitudes, model performance improves with increasing resolution. However, the performance gains from higher resolutions are less pronounced than expected. This could be due to two main factors: first, high-resolution images represent a smaller portion of the training samples in the foundation model; second, the visual encoder's limited output tokens require compression of high-resolution visual data. Figure 5 shows the distribution of the area of pixels occupied by the vehicle under different AGLs, obtained by multiplying the length by the width of

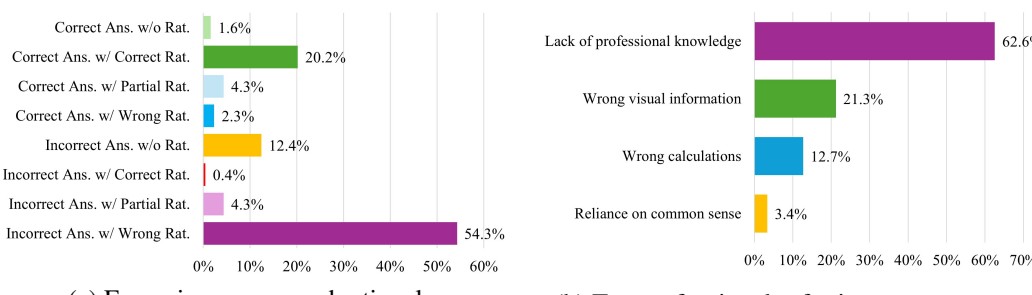

(a) Errors in answers and rationales

(b) Types of rationales for incorrect answers

Figure 6: Error analysis of GPT-4o results. Notations: "Answer" is "Ans.", "Rationale" is "Rat.", "Partially Correct" is "Partial".

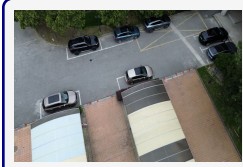

**Question:** There were 5 cars in drone monitoring area two hours ago, how many cars entered this area at least within two hours?

**GPT-4o output:**
There are currently 7 cars in the area. If there were 5 cars two hours ago, then at least 2 cars have entered the area within the last two hours.

Answer: 2

**Correct output: 2**

**Question:** What are the coordinates of the projection point of the point 1 meter above the ground plane at the center of the vehicle in the pixel coordinate system?

**Choices:** (A) [2847, 1986]   (B) [1153, 1986]
(C) [1153, 264]   (D) [2847, 264]

**GPT-4o output:** From the image, the center of the vehicle appears to be around the coordinates [2847, 1986]. Since the point is 1 meter above the ground, the projection point will be displaced upwards by approximately 69 pixels. New y-coordinate = 1986 − 69 ≈ 1917 Therefore, the new coordinates are approximately [2847, 1917]. Thus, the best option is: [2847, 1986]

**Correct output: (A) [2847, 1986]**

(a) **Correct answer** and **explanation**

(b) **Correct answer** but **wrong explanation**

Figure 7: Two examples from GPT-4o. In (b), GPT-4o is unable to accurately estimate the projection point. A precise calculation should take into account the imaging geometry knowledge.

the vehicle's rotated bounding box. When the UAV is at high AGL, vehicles only occupy a smaller area in the images. The visual token obtained after downsampling in the visual encoder will lose some of the detailed features of small targets. We present more examples in §E.3 to further illustrate the impact of image resolution.

**Impact of zero-shot prompting.** We attempt to enhance the reasoning performance of VLM, by applying zero-shot prompting strategies. The CoT method, simply by appending "Let's think step by step", significantly improves performance on InternVL and GPT-4o. Specifically, it improves InternVL2-40B by 3.4 points. However, in LLaVA-v1.6-34B, the performance decreases by 3.2 points. This reflects its deficiency in multi-step reasoning capability. This highlights the disparity between the two models in terms of multi-step reasoning capability. The PS method, based on the "plan-then-execute" approach, improves the performance of GPT-4o by 1.1 points, placing it first among all models. This indicates that GPT-4o has the ability to design a plan for specific problems and can eliminate some of the original errors through this planning process. Recently, several zero-shot prompting methods tailored for VLMs have emerged, such as DCoT and CCoT, but experimental results indicate that their performance remains suboptimal. We attribute this to RS images that contain numerous small objects, making it difficult to fully describe or relate them compared to natural images. More examples of zero-shot reasoning techniques are provided in §E.5.

## 3.4 QUALITATIVE ANALYSIS

**Success and failure analysis of GPT-4o.** In §3.3, GPT-4o is currently the top-ranked model in GEOMATH. To understand its success and failure, we perform a two-stage manual analysis of the model's output. In the first stage, we assessed the correctness of the rationale provided by

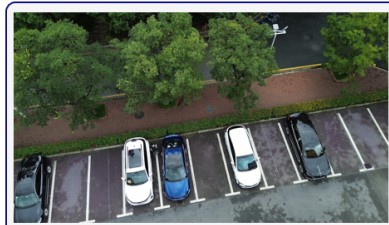

**Question:** How many vehicles are priced above 100000 RMB in this image?
**Ground truth:**
- black audi a6l: 430000 RMB,
- white changan cs75: 80000 RMB,
- blue hyundai lafesta: 145000 RMB,
- white buick velite 6: 100000 RMB,
- black hongqi h9: 535000 RMB
**Answer:** 3

**GPT-4o output:**
From the image:
1.The black car on the far left appears to be a luxury sedan.
2.The white car next to it seems to be a mid-range SUV.
3.The blue car in the middle looks like a sports car.
4.The white car next to the blue car appears to be a mid-range sedan.
5.The black car on the far right also appears to be a luxury sedan.
Based on this assessment, it is reasonable to assume that the black cars on the far left and far right, as well as the blue sports car in the middle, are likely priced above 100,000 RMB.
**Answer:** 3

**InterVL2 output:**
There are three vehicles priced above 100,000 RMB in the image. Here is a description of each vehicle:
1.The first vehicle is a white sedan parked in the second parking spot from the left.
2. The second vehicle is a blue sedan parked in the third parking spot from the left.
3. The third vehicle is a white sedan parked in the fourth parking spot from the left.
**Answer:** 3

Figure 8: Comparison of cross-view knowledge transfer ability Between InternVL2 and GPT-4o.

the model and then evaluated the precision of the results based on the answers extracted through regularization. Figure 6 (a) illustrates the eight patterns of GPT-4o outputs judged manually. We find that 54.3% of the outputs are incorrect answers with the wrong rationale, indicating the models' deficiency in reasoning capabilities within the RS domain. Even among the correct answers, there is a 2.3% chance of being accompanied by incorrect rationale. In the second stage, we summarize four common types of reasoning errors through observation: reliance on common sense, lack of domain-specific knowledge, computational errors, and incorrect visual cues. Figure 6 (b) shows the classification of reasons for erroneous rationale. The primary cause of reasoning errors is the model's lack of domain-specific knowledge in remote sensing, which also explains why GEOMATH presents a greater challenge compared to existing multimodal mathematical reasoning benchmarks. The second most common cause is the failure to accurately extract key visual clues, which accounts for 21. 9%, highlighting the model's deficiency in perception capabilities for RS images. We perform a qualitative analysis of representative examples generated by GPT-4o. In Figure 7 (a), we find that GPT-4o not only produces the correct answers but also provides accurate reasoning, including the correct method to calculate cars. However, in Figure 7 (b), while the model predicts the correct answer, it fails to give the correct reasoning. Its logic is correct, but it lacks the imaging geometry to perform precise calculations.

**Comparison of InternVL and GPT-4o.** Interestingly, we observe that GPT-4o demonstrates the ability to infer vehicle prices based on visual attributes observed from an aerial view. As shown in Figure 8, GPT-4o correctly assessed the price of each vehicle, while InternVL2, despite arriving at the correct answer by chance, provided an incorrect analysis. Even for humans, attempting to determine fine-grained details of a vehicle from aerial images is highly challenging. To our knowledge, no existing RS data provides vehicle price information for training, which validates the cross-view knowledge transfer ability of GPT-4o. Further analysis in §E.6 reveals that GPT-4o outperforms other models in answering price-related questions. This suggests that GPT-4o is able to estimate vehicle prices more accurately from an aerial perspective based on existing knowledge. By revealing the potential gap between the two best performing VLMs in GEOMATH, we hope to provide some guidance for future research. More comparisons of various VLMs can be found in §E.7.

## 4 RELATED WORK

Several benchmarks (Lu et al., 2024b; Wang et al., 2024b; Liu et al., 2024b) have been proposed to evaluate the multimodal mathematical reasoning capabilities of VLMs, but most focus on pure mathematical theory and computation, without involving remote sensing expertise. Existing benchmarks, such as MathVista (Lu et al., 2024b), rely primarily on small figures, charts, and few natural images to provide visual context. This work presents a domain-specific multimodal mathematical reasoning benchmark that leverages high-resolution RS images as visual contexts.

The strong performance of LLMs enables VLGFMs to transparently present their entire reasoning process, offering a new pathway to develop trustworthy RS interpretation systems (Wang et al., 2024c). However, existing VLGFM benchmarks (Hu et al., 2023; Li et al., 2024) provide only final answers, omitting intermediate reasoning steps, which hinders the evaluation of the validity of the reasoning and the reliability of the answers (Chen et al., 2024). To address this gap, we introduce the first VLGFM benchmark that incorporates multistep reasoning processes and features longer reasoning steps than existing RS VQA datasets.

## 5 CONCLUSION

In this work, we propose GEOMATH, a novel benchmark designed to evaluate the mathematical reasoning capabilities of VLMs in the context of RS imagery. We evaluated 14 prominent models and observed that even advanced models like GPT-4o struggle due to a lack of domain-specific mathematical knowledge. Furthermore, we highlight the detrimental effect of low-resolution input on model performance, emphasizing that fully utilizing visual clues in high-resolution RS imagery with many small objects is crucial. Moreover, our analysis of the reasons behind GPT-4o's reasoning errors offers valuable insights for future investigations.

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

CONTENTS

# A PROBLEM DESIGN

## A.1 MATHEMATICAL REASONING DEFINITION

Six subjects of mathematical reasoning in remote sensing are defined in Table 3.

| Mathematical Subject | Description |
|---|---|
| Geometry (28.6%) | It emphasizes *spatial* understanding, analysis of 2D and 3D coordinate system, and reasoning about their *relationships*. Measure distance, size, area, and angle based on imaging principles and perspective transformation. |
| Logic (20.3%) | It focuses on *critical thinking*, *induction*, and *deduction* reasoning from provided information. The key components include premises, conclusions, and the use of abstract reasoning. |
| Statistics (17.4%) | It focuses on *data interpretation* and *analysis*, such as measuring the maximum, minimum, median, mean, and mode. |
| Arithmetic (14.6%) | It covers the *fundamental operations* such as addition, subtraction, multiplication, and division. |
| Counting (9.5%) | It involves determining the number of specific objects based on single or multiple constraints. |
| Algebra (9.5%) | It encompasses understanding *variables*, *equations*, such as solving univariate and multivariate equations. |

Table 3: Definitions and proportions of six mathematical subjects in GEOMATH.

## A.2 MATHEMATICAL REASONING EXAMPLES

| Math | Examples |
|------|----------|
| GEO | 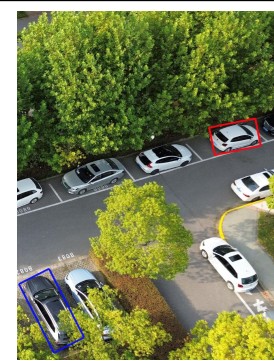 **Context:** The sensor parameters that may be used are as follows: Focal Length: 12 millimeters. Pixel Size: 0.004325 millimeters. Image Width: 4000 pixels. Image Height: 2250 pixels. **Question:** How many meters are the two vehicles in the red and blue boxes apart? **Rationale:** Step 1, locate the center point of two vehicles in the red and blue boxes: [3554, 1051] and [2583, 1974] Step 2, convert them from pixel to image coordinate system: ... Step 3, convert them from image to camera coordinate system: ... Step 4, calculate the distance in the camera coordinate system: $\sqrt{(24.4-6.8)^2+(-1.2-9.9)^2+(43.6-32.5)^2} \approx 23$ **Answer: 23** |
| LOG | 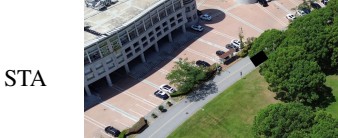 **Question:** There were 27 cars in this area an hour ago, how many cars have entered this area at least within an hour? **Solution:** Step 1, count all current vehicles: 29 Step 2, the number of cars entering the area is at least equal to the increase in the number of cars in this area: 29-27=2 **Answer: 2** |
| STA | 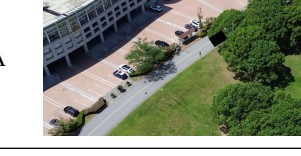 **Question:** What color of vehicle is most common in the image? **Rationale:** Step 1, identify the color of all vehicles: ['white', 'brown', ...] Step 2, count vehicles for each color: {'white': 7, 'brown': 3, ...} Step 3, sort to get the most common color: white **Answer: white** |
| ARI | **Context:** The vehicle price dictionary that may be used is as follows: {'nio ec6': 385000, 'byd dolphin': ...} **Question:** What is the price difference between the car in the red box and the car in the blue car? (Unit: RMB) **Rationale:** Step 1, identify the model of two cars: byd song plus and aito m5 Step 2, query the prices of two vehicles: 155000 and 265000 Step 3, calculate the price difference: 265000-155000=110000 **Answer: 110000** |
| CNT | 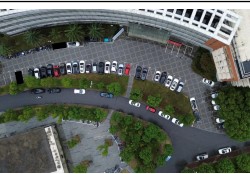 **Question:** How many SUV vehicles are there in the image? **Rationale**: Step 1, identify the type of all vehicles: ['suv', 'suv', ...] Step 2, count all SUV vehicles: 17 **Answer: 17** |
| ALG | 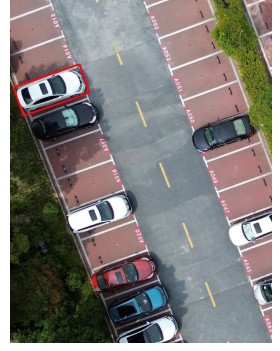 **Context:** The sensor parameters that may be used are as follows: Focal Length: 12 millimeters. Pixel Size: 0.004325 millimeters. Image Width: 4000 pixels. Image Height: 2250 pixels. **Question:** The equation of the ground plane in the camera coordinate system is: -cos(90)*y-sin(90)*z+40=0. What are the coordinates of the center point of the vehicle in the red box in the camera coordinate system? (Unit: meter) **Rationale:** Step 1, locate the center point of the vehicle: [420, 534] Step 2, convert the center point of the vehicle from the pixel coordinate system to the image coordinate system: [-6, -2] Step 3, convert the center point of the vehicle from the image coordinate system to the camera coordinate system: [-22, -8, 40] **Answer: [-22, -8, 40]** |

Table 4: Examples of six mathematical reasoning subjects in GEOMATH.

## A.3 TOPIC SUMMARY

The topics are summarized in Table 5.

| Topic | Subject | Visual Skill | Application |
|---|---|---|---|
| Perspective Geometry | GEO | Location | Surveying |
| Metric Geometry | GEO | Location | Surveying & Military |
| Spatial Relation | GEO | FG Recognition, Location | Surveying & Military |
| Comparison | LOG | FG Recognition, Visual Prompt | Entertainment |
| Deduction | LOG | FG Recognition, Visual Prompt | Surveillance |
| Induction | LOG | FG Recognition | Surveillance |
| Maximum | STA | FG Recognition, Location | Market Research |
| Minimum | STA | FG Recognition, Location | Market Research |
| Mean | STA | FG Recognition, Location | Market Research |
| Median | STA | FG Recognition, Location | Market Research |
| Mode | STA | FG Recognition, Location | Market Research |
| Addition | ARI | FG Recognition | Market Research |
| Subtraction | ARI | FG Recognition, Visual Prompt | Market Research |
| Multiplication | ARI | FG Recognition, Visual Prompt | Market Research |
| Division | ARI | FG Recognition, Visual Prompt | Market Research |
| Counting based on single property | CNT | FG Recognition | Market Research |
| Counting based on multiple property | CNT | FG Recognition | Market Research |
| Counting based on comparison | CNT | FG Recognition | Market Research |
| Univariate Equation | ALG | Location, Visual Prompt | Surveying |
| Multivariate Equations | ALG | Location, Visual Prompt | Surveying |

Table 5: Summary of the 20 different topics in GEOMATH. The table provides details on their subject and visual skill types. **Location** represents the ability to provide the pixel coordinates of key points. **FG recognition**, short for fine-grained recognition, refers to the ability to identify critical visual cues in RS images, including the specific properties and models of vehicles. **Visual prompt** indicates the capability to determine the referenced target based on various colored boxes added to the image. **Surveying** suggests that remote sensing professionals can leverage this capability to enhance the efficiency of geological surveys and obtain interpretable and reliable results. **Military** indicates that it can be used in unmanned warfare to improve the intelligence level of drones. **Entertainment** indicates that users can utilize this capability to satisfy their curiosity. **Surveillance** indicates that this capability can be used to monitor activities within a specific area. **Market research** indicates that automotive companies can leverage this capability to conduct fine-grained analysis of customer preferences within a specific region.

# B  DATA COLLECTION DETAILS

## B.1  UAV DATA COLLECTION INFORMATION

| Scenario | Date | Time | Weather | AGL | | | | | | | | | Pitch Angle | | |
|---|---|---|---|---|---|---|---|---|---|---|---|---|---|---|---|
| | | | | 20 | 30 | 40 | 60 | 70 | 80 | 100 | 110 | 120 | 45 | 60 | 90 |
| A | 0910 | Noon | Sunny | 17 | 7 | 10 | 6 | 9 | 5 | 4 | 8 | 5 | 24 | 27 | 20 |
| B | 0911 | Noon | Sunny | 9 | 7 | 4 | 2 | 4 | 3 | 2 | 2 | 2 | 13 | 13 | 9 |
| C | 0912 | Morning | Sunny | 12 | 14 | 8 | 6 | 8 | 4 | 3 | 7 | 3 | 29 | 21 | 15 |
| D | 0912 | Afternoon | Sunny | 13 | 10 | 7 | 5 | 7 | 4 | 2 | 5 | 1 | 22 | 20 | 12 |
| E | 0913 | Morning | Cloudy | 11 | 7 | 4 | 4 | 4 | 3 | 4 | 4 | 2 | 15 | 19 | 9 |
| F | 0913 | Morning | Cloudy | 7 | 15 | 6 | 9 | 6 | 4 | 10 | 1 | 2 | 22 | 26 | 12 |
| G | 0914 | Noon | Cloudy | 13 | 6 | 3 | 3 | 4 | 3 | 5 | 6 | 3 | 16 | 21 | 9 |
| H | 0914 | Noon | Cloudy | 8 | 5 | 6 | 4 | 3 | 3 | 4 | 9 | 2 | 17 | 16 | 11 |
| I | 0915 | Noon | Rainy | 13 | 15 | 11 | 5 | 8 | 3 | 3 | 7 | 2 | 30 | 21 | 16 |
| J | 0915 | Afternoon | Cloudy | 18 | 14 | 10 | 7 | 4 | 7 | 6 | 3 | 3 | 21 | 31 | 20 |
| K | 0916 | Noon | Cloudy | 11 | 11 | 7 | 10 | 4 | 4 | 4 | 13 | 4 | 28 | 25 | 15 |

Table 6: The data collected by the drone covers multiple weather conditions, AGLs, and pitch angles.

## B.2  DETAILS OF MATADATA

| Type | Details |
|---|---|
| Camera parameters | Focal length, ISO, pixel size, shutter speed, aperture, sensor size, image resolution, pitch angle, AGL, latitude, longitude, timestamp. |
| Vehicle attributes | Location of pixel coordinate system, rotated bounding box, front direction, brand, model, color, type, powertrain, length, width, height, sunroof, roof rack, max price, min price, number of doors / seats. |

Table 7: Details of metadata, where most vehicle attributes are obtained from the ground video.

## B.3  DETAILS OF COORDINATE SYSTEM TRANSFORMATION

The complete derivation processes for two coordinate system transformations are provided here.

The transformation between the pixel coordinate system and the image coordinate system can be represented by an affine matrix, as follows:

$$\begin{bmatrix} x_P \\ y_P \\ 1 \end{bmatrix} = \begin{bmatrix} \frac{1}{p} & 0 & \frac{w}{2} \\ 0 & \frac{1}{p} & \frac{h}{2} \\ 0 & 0 & 1 \end{bmatrix} \begin{bmatrix} x_I \\ y_I \\ 1 \end{bmatrix} \tag{1}$$

where $p$ represents the pixel size of sensor. $\frac{w}{2}$ and $\frac{h}{2}$ denote the origin offsets, with the origin of the image coordinate system typically located at the image's top-left corner. Given the pixel coordinates of a certain point, its corresponding image coordinates can be calculated as follows:

$$\begin{cases} x_I = (x_P - w/2) \cdot p \\ y_I = (y_P - h/2) \cdot p \end{cases} \tag{2}$$

The transformation from the camera coordinate system to the image coordinate system is a conversion from three-dimensional to two-dimensional coordinates. Assuming the focal length of the camera is $f$, then we have

$$z_c \begin{bmatrix} x_I \\ y_I \\ 1 \end{bmatrix} = \begin{bmatrix} f & 0 & 0 & 0 \\ 0 & f & 0 & 0 \\ 0 & 0 & 1 & 0 \end{bmatrix} \begin{bmatrix} x_C \\ y_C \\ z_C \\ 1 \end{bmatrix} \tag{3}$$

where $z_C$ denotes the depth of the point, which can be obtained by a depth camera (binocular or structured light). Because the drone camera we are using cannot provide depth information, we need to find another way.

When the ground satisfies the ground plane assumption, given the AGL of the drone and the pitch angle of the camera, the ground plane equation in the camera coordinate system is as follows:

$$-\cos\theta \cdot Y_C - \sin\theta \cdot Z_C + H = 0 \tag{4}$$

The equation of the line connecting the camera origin to the projection point on the pixel plane in the camera coordinate system is given by:

$$\begin{cases} X_C = x_I \cdot t \\ Y_C = y_I \cdot t \\ Z_C = f \cdot t \end{cases} \tag{5}$$

Substituting the line equation into the ground plane equation yields:

$$t = \frac{H}{y_I \cos\theta + f \sin\theta} \tag{6}$$

Substituting $t$ back into the line equation yields:

$$(\frac{x_I H}{y_I \cos\theta + f \sin\theta}, \frac{y_I H}{y_I \cos\theta + f \sin\theta}, \frac{f H}{y_I \cos\theta + f \sin\theta}) \tag{7}$$

To preserve the spatial mapping between camera coordinates and pixel coordinates, we refrained from cropping the 4K images to increase the dataset size, as is commonly done in most remote sensing datasets.

## C   MORE DATASET ANALYSIS

| Dataset | Images | | #VQAs | Mathematical Subject | | | | | | Rationale |
| | Number | Size | | CNT | GEO | LOG | ARI | ALG | STA | |
| --- | --- | --- | --- | --- | --- | --- | --- | --- | --- | --- |
| RSVQA-LR (Lobry et al., 2020) | 772 | 512 | 77,232 | ✓ | ✓ | ✗ | ✗ | ✗ | ✗ | ✗ |
| RSVQA-HR (Lobry et al., 2020) | 100,659 | 512 | 1,066,316 | ✓ | ✓ | ✗ | ✗ | ✗ | ✗ | ✗ |
| RSVQAxBEN (Lobry et al., 2021) | 590,325 | 20 to 120 | 14,758,150 | ✗ | ✗ | ✗ | ✗ | ✗ | ✗ | ✗ |
| FloodNet (Rahnemoonfar et al., 2021) | 4,056 | **4,000** | 11,000 | ✓ | ✗ | ✗ | ✗ | ✗ | ✗ | ✗ |
| RSIVQA (Zheng et al., 2021) | 37,264 | 256 to 4,000 | 111,134 | ✓ | ✓ | ✗ | ✗ | ✗ | ✗ | ✗ |
| CDVQA (Yuan et al., 2022) | 2,968 | 512 | 122,000 | ✗ | ✗ | ✗ | ✓ | ✗ | ✗ | ✗ |
| VQA-TextRS (Al Rahhal et al., 2022) | 2144 | 256 to 600 | 6245 | ✗ | ✓ | ✗ | ✗ | ✗ | ✗ | ✗ |
| CRSVQA (Zhang et al., 2023a) | 4,639 | 600 | 4,644 | ✓ | ✓ | ✗ | ✗ | ✗ | ✗ | ✗ |
| RSIEval (Hu et al., 2023) | 100 | 512 | 936 | ✓ | ✓ | ✗ | ✗ | ✗ | ✗ | ✗ |
| EarthVQA (Wang et al., 2024a) | 6,000 | 1024 | 208,593 | ✓ | ✓ | ✗ | ✗ | ✗ | ✗ | ✗ |
| VRSBench (Li et al., 2024) | 29,614 | 512 | 123,221 | ✓ | ✓ | ✗ | ✗ | ✗ | ✗ | ✗ |
| GEOMATH | 360 | **4,000** | 3,773 | ✓ | ✓ | ✓ | ✓ | ✓ | ✓ | ✓ |

Table 8: Comparison between existing remote sensing vision-language datasets and our GEOMATH dataset. GEOMATH dataset provides a more comprehensive coverage of mathematical problems. Additionally, it is the first RS VQA dataset to provide the rationale, which means reasoning processes.

Previous datasets offer counting-type VQAs based on a single condition, with the object attributes being relatively few and primarily focused on color. GEOMATH not only enriches the attributes of the object, but also introduces object counting under multiple constraints, significantly increasing the difficulty. Moreover, GEOMATH is the first to extend spatial relationships from the plane to three-dimensional space, substantially enhancing the complexity of tasks, while previous datasets provided geometric problems that were restricted to planar spatial relationships.

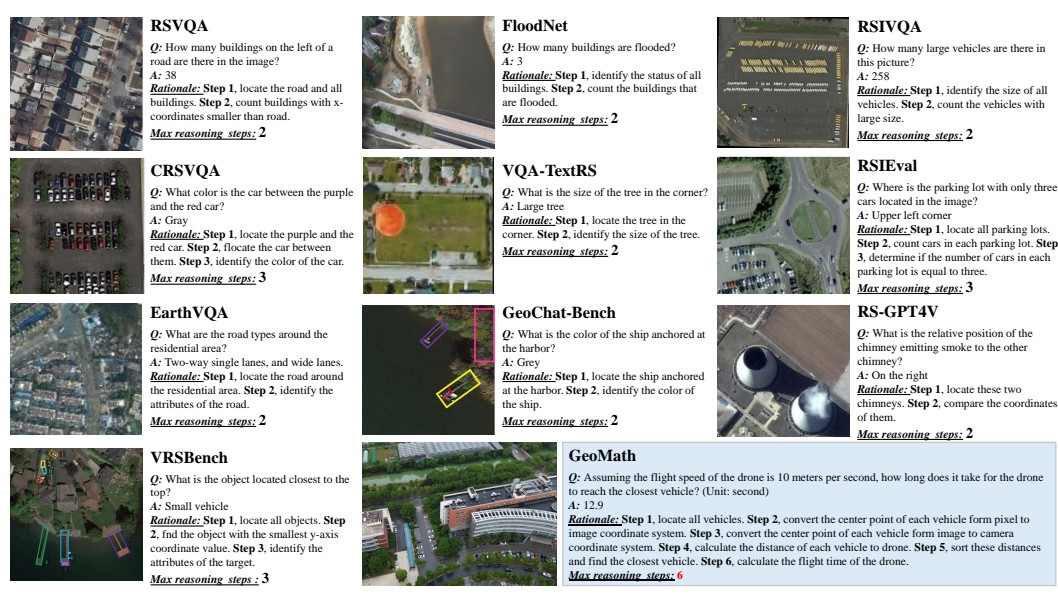

Figure 9: Examples of mathematical problems requiring the maximum reasoning steps across various RS VQA benchmarks. Except for GeoMath, these benchmarks do not explicitly provide reasoning steps; the examples shown are manual analysis results. Undoubtedly, GEOMATH currently has the longest reasoning steps among RS VQA benchmarks.

# D   MORE DETAILS ON THE SETUP

## D.1   PROMPTS FOR RESPONSE GENERATION

The prompt used to instruct the foundation models to generate responses is illustrated in Table 9.

| Question type | Stage | Task instruction |
|---|---|---|
| Multiple-choice | Generation | Observe this image captured by a drone and answer the question by choosing the best option. Question: {question} Choices: {choices} |
| Multiple-choice | Extraction | Based on the question ({question}) and reasoning provided in the output, conclude the final answer in the format 'Answer: $LETTER' (without quotes) where LETTER is one of ABCD. |
| True/False | Generation | Observe this image captured by a drone and answer the question. Question: {question} |
| True/False | Extraction | Based on the question ({question}) and reasoning provided in the output, conclude the final answer in the format 'Answer: Yes' or 'Answer: No' (without quotes). |
| Free-form | Generation | Observe this image captured by a drone and answer the question. Question: {question} |
| Free-form | Extraction | Based on the question ({question}) and reasoning provided in the output, conclude the final answer in the format 'Answer: XX' (without quotes). |

Table 9: The task instructions for different question types.

## D.2   MODEL HYPERPARAMETERS

The hyperparameters for the experiments in §3.2 are set to their default values unless otherwise specified. Table 10 details the specific generation parameters for the various VLMs we evaluated.

| Model | Generation Setup |
|---|---|
| GPT-4o | Official API, model = `gpt-4o`, temperature = 0, max_tokens = 1000, evaluation dates range from Sep 12 to 18, 2024. |
| GeoChat | do_sample = False, temperature = 0.0, max_new_tokens = 1000 |
| Others | Framework: https://github.com/InternLM/lmdeploy session_len = 8192, temperature = 0.0, max_tokens = 1000 |

Table 10: Generating parameters for various VLMs.

# E MORE EXPERIMENTAL RESULTS

## E.1 ANALYSIS OF PITCH ANGLE

Among the 14 models, we find that half the models, represented by GPT-4o, achieve the highest scores at a camera pitch angle of 90 degrees, while other half models, e.g., InternVL2-40B, perform better at 60 degrees. All models show the poorest performance at a pitch angle of 45 degrees, which can be attributed to the lack of low-angle samples during training. Therefore, enhancing the generalizability of RS VLMs under different viewing angles is a potential research direction.

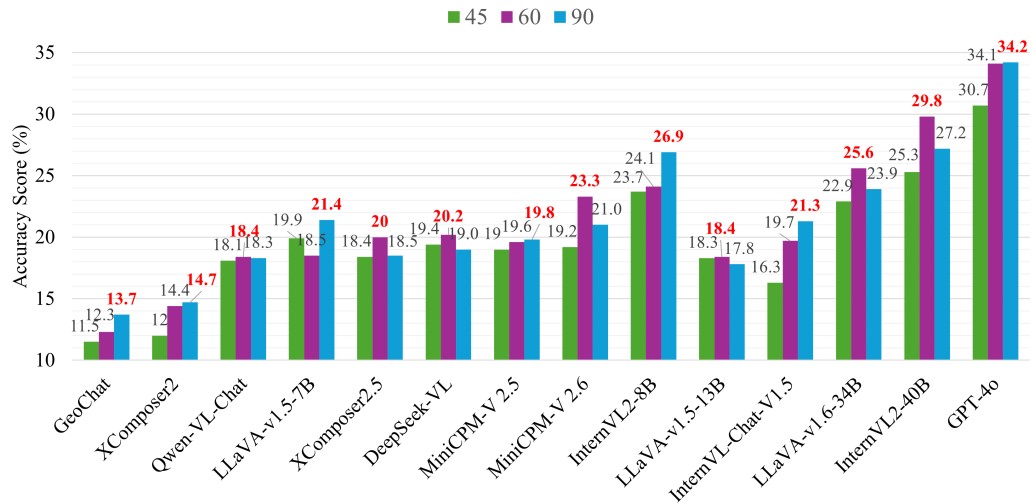

Figure 10: Impact of camera pitch angle on model performance.

## E.2 ANALYSIS OF RESPONSE LENGTH

We analyze the accuracy scores and average response lengths of 14 models and find a clear positive correlation between them. The longer the response lengths of the model, the more likely it is to achieve higher accuracy on GEOMATH.

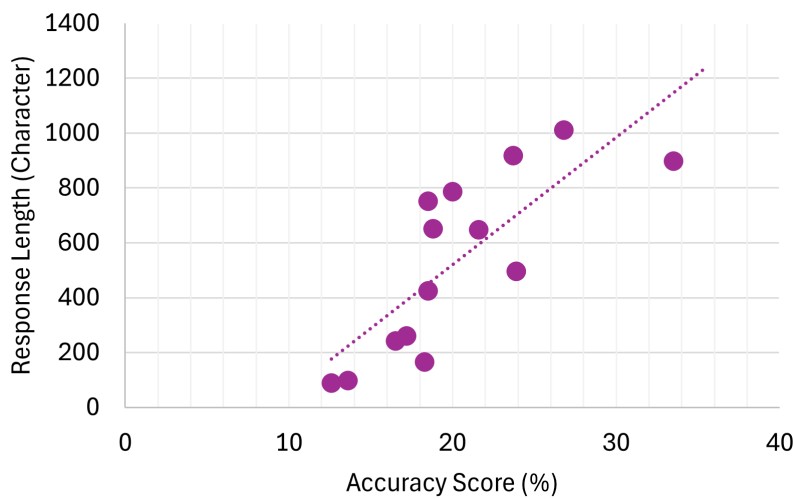

Figure 11: The relationship between the model's response length and accuracy score.

### E.3  IMPACT OF IMAGE RESOLUTION

Figure 12 illustrates the detrimental impact of lower image resolution on InternVL2-40B. The AGL of this photo is 100 meters and the pitch angle is 60 degrees. As the image resolution decreases from 4000×2250 to 336×189, the number of pixels representing the car in the red box drops from 6926 to just 44 (calculated based on the length and width of the rotated bounding box), causing the model to lose its ability to accurately recognize vehicle types. Earlier VLMs typically used fixed input resolutions (Liu et al., 2023), often with significantly lower values. To fully exploit the visual clues available in the GEOMATH images, enabling support for 4K input resolution would be a promising direction for future research on VLM.

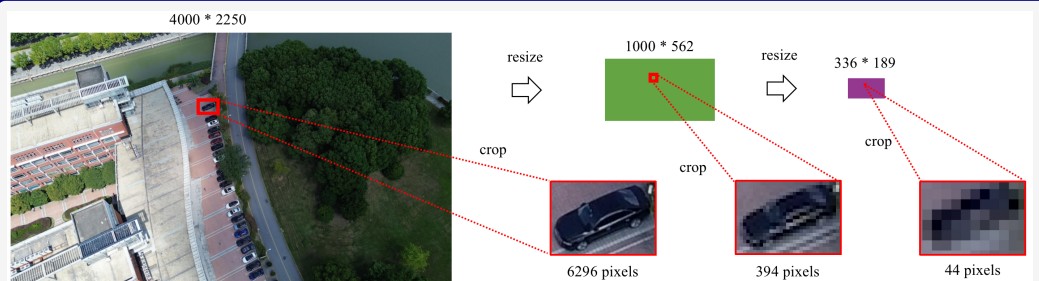

**Question:** How many mid-size vehicles are there in the image?

**Output when image resolution equals to 4000×2250:**
To determine the number of mid-size vehicles in the image, let's carefully examine the parking area. 1. **Identify the Vehicles**: Look for vehicles that are neither too small (such as compact cars) nor too large (such as SUVs or trucks). Mid-size vehicles typically include sedans and some small SUVs. 2. **Count the Vehicles**: The parking area has a total of 18 vehicles. By visually inspecting each vehicle, we can estimate its size. 3. **Estimate Mid-Size Vehicles**: Upon close inspection, it appears that 10 vehicles are mid-size. Therefore, based on visual inspection, there are 10 mid-size vehicles in the image.
**Answer: 10**

**Output when image resolution equals to 1000×562:**
To determine the number of mid-size vehicles in the image, let's carefully examine the parking area. 1. **Identify the Vehicles**: Look for vehicles that are neither too small (such as compact cars) nor too large (such as SUVs or trucks). Mid-size vehicles typically include sedans and some small SUVs. 2. **Count the Vehicles**: Starting from the left, there are several rows of parked vehicles. In the first row, there are two vehicles that appear to be mid-size. In the second row, there are three vehicles that appear to be mid-size. In the third row, there are two vehicles that appear to be mid-size. In the fourth row, there are three vehicles that appear to be mid-size. 3. **Summarize the Count**: First row: 2 mid-size vehicles. Second row: 3 mid-size vehicles. Third row: 2 mid-size vehicles. Fourth row: 3 mid-size vehicles 4. **Total Count**: 2 + 3 + 2 + 3 = 10. Therefore, there are 10 mid-size vehicles in the image.
**Answer: 10**

**Output when image resolution equals to 336×189:**
To determine the number of mid-size vehicles in the image, let's carefully examine the scene. The image shows a parking lot adjacent to a building and a road. The vehicles are parked in a row, and we need to identify the mid-size vehicles among them. Mid-size vehicles typically have a length between 180 inches (4.57 meters) and 200 inches (5.08 meters). They are larger than compact cars but smaller than full-size vehicles. Upon close inspection, it appears that all the vehicles in the parking lot are mid-size vehicles. They are consistent in size and shape, fitting the typical characteristics of mid-size cars. Therefore, based on common knowledge and information available in the image, I estimate that there are approximately 15 mid-size vehicles.
**Answer: 15**

Figure 12: Example illustrating the impact of image resolution on the InternVL2-40B. As the resolution decreases, reasoning vehicle types from image information becomes increasingly challenging.

### E.4 ANALYSIS OF GEOCHAT

We found that fine-tuning models using task-specific RS image-text instruction sets may impair their capabilities on some multimodal reasoning tasks. This indicates that existing VLGFMs are fundamentally still engaged in data fitting, which does not lead to the emergence of true RS intelligence. This raises an important question. What kind of image-text instruction set can lead us toward a "GPT-4v moment" in remote sensing?

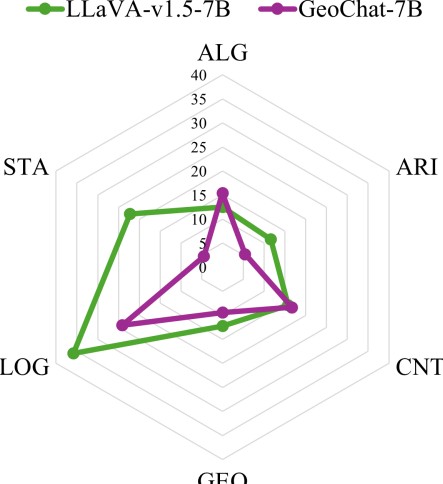

Figure 13: Performance difference analysis between LLaVA-v1.5-7B and GeoChat-7B. The latter can be roughly regarded as the former finetuned on RS data.

## E.5 MORE EXAMPLES OF ZERO-SHOT PROMPTING TECHNIQUES

**Question:** Assuming the flight speed of the drone is 16 meters per second, how long does it take for the drone to reach the closest vehicle? (Unit: second)

**Ground truth: 9.4**

**Solution:**

Step 1, locate the center point of each vehicle. Step 2, convert center point of each vehicle from pixel coordinate system to image coordinate system. Step 3, convert center point of each vehicle from image coordinate system to camera coordinate system. Step 4, calculate distance of each vehicle in camera coordinate system. Step 5, find the minimum distance to the drone: 150.9. Step 6, calculate the flight time of the drone: $150.9/16 = 9.4$

**Model Response:**

To determine the time it takes for the drone to reach the farthest vehicle, we need to calculate the distance it needs to travel and then divide that by the drone's speed.

1. Determine the distance to the farthest vehicle:
- The farthest vehicle is located at the edge of the parking lot, which is horizontally aligned with the drone's position.
- The distance can be approximated by considering the width of the parking lot and the distance from the drone's position to the farthest vehicle.

2. Estimate the width of the parking lot:
- The parking lot appears to be approximately 100 meters wide.

3. Calculate the distance to the farthest vehicle:
- The drone is positioned at an altitude of 120 meters.
- The farthest vehicle is at the edge of the parking lot, which is approximately 100 meters horizontally from the drone's position.
- Therefore, the total horizontal distance to the farthest vehicle is 120 meters (altitude) + 100 meters (horizontal distance) = 220 meters.

4. Calculate the time taken:
- The drone's speed is 16 meters per second.
- Time = Distance / Speed
- Time = 220 meters / 16 meters per second
- Time = 13.75 seconds

So, it takes approximately 13.75 seconds for the drone to reach the farthest vehicle.

Figure 14: The response of the InternVL2-40B model without CoT.

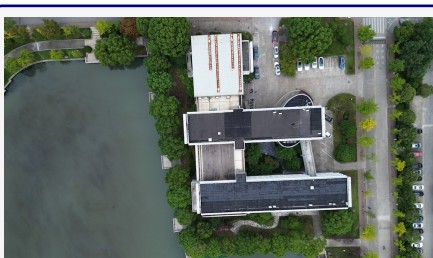

**Question:** Assuming the flight speed of the drone is 16 meters per second, how long does it take for the drone to reach the closest vehicle? (Unit: second)
**Ground truth: 9.4**
**Solution:**
Step 1, locate the center point of each vehicle. Step 2, convert center point of each vehicle from pixel coordinate system to image coordinate system. Step 3, convert center point of each vehicle from image coordinate system to camera coordinate system. Step 4, calculate distance of each vehicle in camera coordinate system. Step 5, find the minimum distance to the drone: 150.9. Step 6, calculate the flight time of the drone: $150.9/16 = 9.4$

**Model Response:**
To determine the time it takes for the drone to reach the farthest vehicle, we need to calculate the distance it needs to travel and then divide that by the drone's speed.

1. Determine the distance to the farthest vehicle:
- The farthest vehicle is located at the edge of the parking lot, which is horizontally aligned with the drone's position.
- The distance can be approximated by considering the width of the parking lot and the distance from the drone's position to the farthest vehicle.

2. Estimate the width of the parking lot:
- The parking lot appears to be approximately 100 meters wide.

3. Calculate the distance to the farthest vehicle:
- The drone is positioned at an altitude of 120 meters.
- The farthest vehicle is at the edge of the parking lot, which is approximately 100 meters horizontally from the drone's position.
- Therefore, the total distance to the farthest vehicle is the sum of the horizontal distance (100 meters) and the vertical distance (120 meters).

4. Calculate the total distance:
$$\text{Total Distance} = \sqrt{(100\,\text{m})^2 + (120\,\text{m})^2} = \sqrt{10000 + 14400} = \sqrt{24400} \approx 156.2\,\text{meters}$$

5. **Calculate the time taken:**
- The drone's speed is 16 meters per second.
- Time taken = Distance / Speed
$$\text{Time} = \frac{156.2\,\text{meters}}{16\,\text{meters/second}} \approx 9.76\,\text{seconds}$$

Therefore, it takes approximately 9.76 seconds for the drone to reach the farthest vehicle.

Figure 15: The response of the InternVL2-40B model based on CoT. The answer is very close.

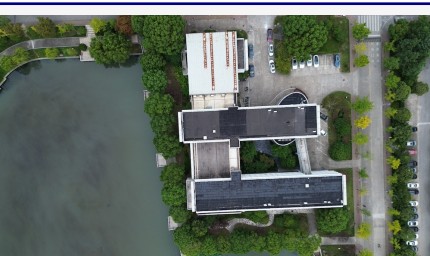

**Question:** Assuming the flight speed of the drone is 16 meters per second, how long does it take for the drone to reach the closest vehicle? (Unit: second)

**Ground truth: 9.4**

**Solution:**
Step 1, locate the center point of each vehicle. Step 2, convert center point of each vehicle from pixel coordinate system to image coordinate system. Step 3, convert center point of each vehicle from image coordinate system to camera coordinate system. Step 4, calculate distance of each vehicle in camera coordinate system. Step 5, find the minimum distance to the drone: 150.9. Step 6, calculate the flight time of the drone: $150.9/16 = 9.4$

**Model Response:**
To determine how long it takes for the drone to reach the farthest vehicle, we need to calculate the distance between the drone and the farthest vehicle and then divide that distance by the drone's speed.

### Step-by-Step Solution:

1. Determine the distance from the drone to the farthest vehicle:
- The image is captured from a height of 120 meters.
- The farthest vehicle is located at the edge of the parking lot, which is approximately 1/3 of the image width from the center.
- The image width is 4000 pixels.
- The pixel size is 0.004325 millimeters.

First, convert the image width from pixels to meters:
Image Width (meters) $= 4000$ pixels $\times 0.004325$ mm/pixel $= 17.3$ meters

Next, calculate the distance from the center of the image to the farthest vehicle:
Distance to farthest vehicle $= \frac{17.3 \,\text{meters}}{2} = 8.65$ meters

2. Calculate the time taken by the drone to reach the farthest vehicle:
- The drone's speed is 16 meters per second.

Now, divide the distance by the speed:
Time $= \frac{\text{Distance}}{\text{Speed}} = \frac{8.65 \,\text{meters}}{16 \,\text{meters/second}} = 0.540625$ seconds

### Conclusion:
The drone takes approximately 0.540625 seconds to reach the farthest vehicle.

This calculation assumes a straight-line distance from the drone to the farthest vehicle and does not account for any potential obstacles or changes in flight path.

Figure 16: The response of the InternVL2-40B model based on PS.

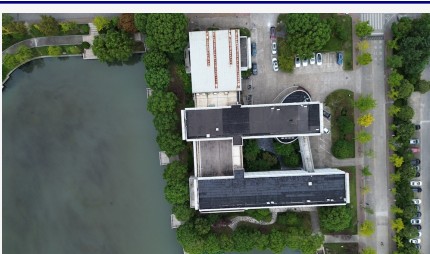

**Question:** Assuming the flight speed of the drone is 16 meters per second, how long does it take for the drone to reach the closest vehicle? (Unit: second)
**Ground truth: 9.4**
**Solution:**
Step 1, locate the center point of each vehicle. Step 2, convert center point of each vehicle from pixel coordinate system to image coordinate system. Step 3, convert center point of each vehicle from image coordinate system to camera coordinate system. Step 4, calculate distance of each vehicle in camera coordinate system. Step 5, find the minimum distance to the drone: 150.9. Step 6, calculate the flight time of the drone: $150.9/16 = 9.4$

**Model Response:**
To determine how long it takes for the drone to reach the farthest vehicle, we need to calculate the distance between the drone and the farthest vehicle and then divide that distance by the drone's speed.

### Relevant Image Information:
1. Drone's Starting Point: The drone is positioned at the top center of the image.
2. Farthest Vehicle: The farthest vehicle is located at the bottom right corner of the image, near the parking lot.
3. Drone's Speed: The drone's speed is 16 meters per second.

### Steps to Calculate the Time:
1. Estimate the Distance:
- The drone appears to be at a height above the farthest vehicle.
- The distance from the drone to the farthest vehicle can be approximated by the diagonal distance from the top center to the bottom right of the image.

2. Approximate Distance Calculation:
- The image dimensions are not provided, but we can estimate the distance by considering the relative sizes of the objects in the image.
- Let's assume the image dimensions are approximately 100 meters by 100 meters (this is a rough estimate based on the size of the buildings and the parking lot).
- The diagonal distance in a square with side length $(s)$ is given by $(s\sqrt{2})$.

3. Calculate the Distance:
- $d \approx 100 \times \sqrt{2} \approx 141.4$ meters.

4. Calculate the Time:

- $t = \frac{\text{Distance}}{\text{Speed}} = \frac{141.4 \text{ meters}}{16 \text{ meters per second}}$
$approx 8.84$ seconds.

### Conclusion:
The drone takes approximately 8.84 seconds to reach the farthest vehicle, assuming the drone's speed is 16 meters per second and the image dimensions are roughly 100 meters by 100 meters.

Figure 17: The response of the InternVL2-40B model based on DCoT.

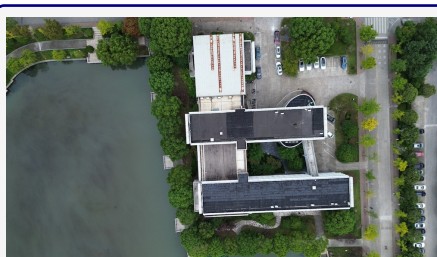

**Question:** Assuming the flight speed of the drone is 16 meters per second, how long does it take for the drone to reach the closest vehicle? (Unit: second)

**Ground truth: 9.4**

**Solution:**

Step 1, locate the center point of each vehicle. Step 2, convert center point of each vehicle from pixel coordinate system to image coordinate system. Step 3, convert center point of each vehicle from image coordinate system to camera coordinate system. Step 4, calculate distance of each vehicle in camera coordinate system. Step 5, find the minimum distance to the drone: 150.9. Step 6, calculate the flight time of the drone: $150.9/16 = 9.4$

**Model Response:**

To answer the question, we need to determine the distance between the drone and the farthest vehicle and then calculate the time it takes for the drone to cover that distance at the given speed.

1. Objects and Attributes:
- Drone: The drone is the object of interest with a specified speed of 16 meters per second.
- Vehicle: The farthest vehicle is the object we need to reach.
- Building: The building is near the drone.
- Parking Lot: The parking lot is where the farthest vehicle is located.
- Trees: Trees surround the building and parking lot.
- Path: A path connects the parking lot to the building.

2. Object Relationships:
- The drone is near the building.
- The farthest vehicle is in the parking lot.
- The building is adjacent to the parking lot.
- Trees surround both the building and the parking lot.
- A path connects the parking lot to the building.

### Step-by-Step Calculation

1. Estimate the Distance:
- From the image, visually estimate the distance between the drone and the farthest vehicle.
- Let's assume the distance is approximately 50 meters.

2. Calculate Time:
- Time = Distance / Speed
- Time = 50 meters / 16 meters per second
- Time = 3.125 seconds

Therefore, it takes approximately 3.125 seconds for the drone to reach the farthest vehicle.

Figure 18: The response of the InternVL2-40B model based on CCoT.

### E.6    CROSS-VIEW KNOWLEDGE TRANSFER ABILITY OF GPT-4O

Previous RS VQA datasets focused primarily on obvious attributes such as color and relative size, and do not require domain-specific automotive knowledge to answer. In contrast, our benchmark captures a variety of fine-grained attributes of ground vehicles, including color, model, brand, price, power type, etc., by aligning drone-captured aerial images with corresponding ground videos. Unlike the default experimental setting, we do not provide any context on vehicle prices. We further filter price-related questions from GEOMATH to quantitatively assess the models' ability to estimate car prices based on the visual appearance of cars in remote sensing images. This specific capability partially reflects the models' cross-view knowledge transfer ability. As expected, GPT-4o achieves an accuracy of 53.2% on price-related questions, demonstrating a dominant performance. InternVL2-40B also performs well, trailing GPT-4o by 12.9 points but still significantly outperforming other models.

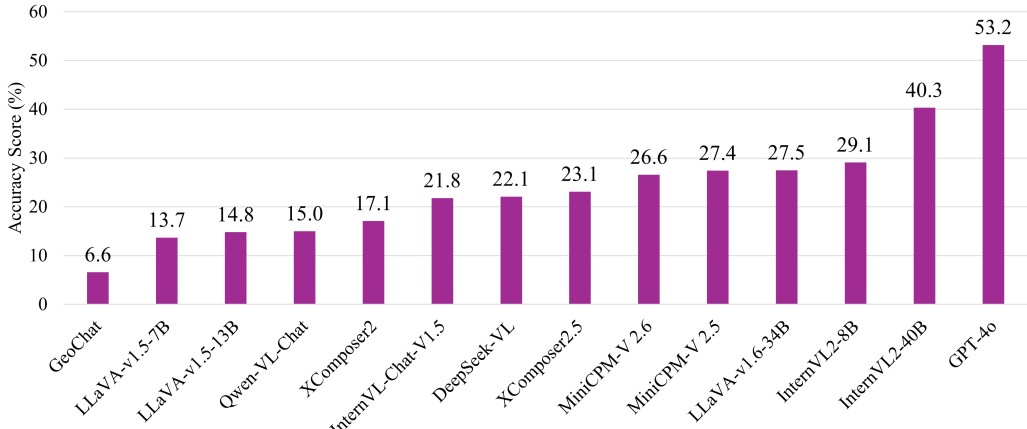

Figure 19: Accuracy scores of VLMs on price-related questions.

E.7   COMPARISONS OF DIFFERENT MODELS

**Question:** What color of SUV vehicles is most common in the image?
**Choices:** A) black. B) pink. C) orange. D) blue.
**Ground truth: D) blue**
**Solution:** Step 1, identify the type of all vehicles.
Step 2, identify the color of suv vehicles.
Step 3, count suv vehicles for each color: {'gray': 2, 'blue': 3, 'brown': 1, 'black': 2}
Step 4, sort to get the most common color of suv vehicle: blue

**XComposer2:** The answer is A.
Answer: A

**GeoChat:** A) black.
Answer: A

**LLaVA-v1.5-7B:** A) black.
Answer: A

**InternVL-Chat-V1.5:** A) black.
Answer: A

**DeepSeek-VL:** A) black.
Answer: A

**InternVL2-8B:** A) black.
Answer: A

**MiniCPM-V 2.5:** The question pertains to identifying the most common color of SUV vehicles in the image. However, the image provided does not contain any SUV vehicles; it is an aerial view of a building complex with solar panels and surrounding greenery. Therefore, none of the choices (A) black, B) pink, C) orange, or D) blue apply to the content of the image. The question cannot be answered based on the provided image.
Answer: None

**MiniCPM-V 2.6:** The most common color of SUV vehicles in the image is black. This can be determined by observing the parking lot where several SUVs are visible, and the majority of them are black. There are no pink, orange, or blue SUVs that can be clearly identified in the image. Therefore, the best option is A) black.
Answer: A

**LLaVA-v1.6-34B:** A) black.
Answer: A

**InternVL2-40B:** A) black.
Answer: A

**GPT-4o:** A) black.
Answer: A

Figure 20: All VLM baselines predicted the wrong answer to this complex problem.

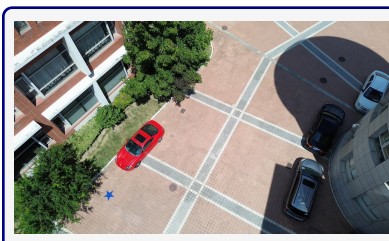

**Question:** How many sports vehicles are there in the image?
**Ground truth: 1**
**Solution:**
Step 1, identify the type of all vehicles: ['sports', 'mpv', 'mid-size', 'subcompact']
Step 2, count all sports vehicles: 1

---

**InternLM-XComposer2:** There are two sports vehicles in the image.
Answer: 2

---

**GeoChat:** There are a total of 2 sports vehicles in the image.
Answer: 1

---

**LLaVA-v1.5-7B:** There are two sports vehicles in the image.
Answer: 2

---

**InternVL-Chat-V1.5:** There are two sports vehicles in the image. One is a red sports car parked on the left side of the parking lot, and the other is a black sports car parked on the right side of the parking lot.
Answer: 2

---

**DeepSeek-VL:** There is one sports vehicle in the image.
Answer: 1

---

**InternVL2-8B:** There are a total of four sports vehicles visible in the image.
Answer: 4

---

**MiniCPM-V 2.5:** In the image, there is one sports vehicle visible. It is a red sports car parked on the brick pavement. The car's design, with its sleek body and low profile, is characteristic of sports cars, which are typically designed for high performance and speed. The other vehicles in the image do not exhibit the same design features and are therefore not classified as sports vehicles.
Answer: 1

---

**MiniCPM-V 2.6:** To determine the number of sports vehicles in the image, one would need to identify vehicles that are typically associated with sports, such as those with a sleek design, low profile, and possibly high-performance features. In this image, the red car in the center appears to be a sports vehicle due to its vibrant color and sporty design. There are no other vehicles in the image that clearly fit the description of a sports vehicle. Therefore, based on the visible evidence, there is one sports vehicle in the image.
Answer: 1

---

**LLaVA-v1.6-34B:** There is one sports vehicle in the image, which is the red car parked on the left side of the street.
Answer: 1

---

**InternVL2-40B:** There is only a single sports vehicle visible in the image.
Answer: 1

---

**GPT-4o:** The image shows one sports vehicle, which is the red car located near the center-left of the image.
Answer: 1

Figure 21: Most VLMs have the ability to identify the type of car from RS images. The MiniCPM series models compare the common features of sports cars with the vehicles in the image during the reasoning process.

