# OpenReview forum: "GeoMath: A Benchmark for Multimodal Mathematical Reasoning in Remote Sensing"
_ICLR.cc/2025/Conference — ICLR 2025 Conference Withdrawn Submission_

### Official Review · Reviewer_oppb · 2024-10-26

**Soundness:** 2
**Presentation:** 3
**Contribution:** 2
**Rating:** 3
**Confidence:** 5

**Summary:**

A new benchmark for mathematical reasoning in remote sensing data is presented.  The authors collect a new dataset using a UVA in Shanghai between Sept 10-16, 2023.  Various VLMs are evaluated on different applications.

**Strengths:**

+ Nicely written paper.
+ Evaluation of various VLMs for math reasoning.
+ Authors develop a detailed evaluation benchmark with new metrics.

**Weaknesses:**

-Despite presenting a detailed benchmark this work is fundamentally limited to data from on particular region.  The conclusions may not transfer well to data from other region.
- For truly developing a benchmark, one needs to look at other remote sensing data such as those from satellite and other sensors.  Only visible images from a UVA are being evaluated.  Perhaps this can be done with surveillance cameras mounted on the top of a building!
- The conclusions are not really clear.  This benchmark doesn't really enhance our understanding of VLMs.
- What are good mitigation approaches to improve the performance of VLMs on this dataset?

**Questions:**

One of the major limitations of this work is the data that is being analyzed.  The data is really not heterogeneous in terms of sensor type, weather conditions, resolution, heights, etc.
Please see my detailed comments above.

---

### Official Review · Reviewer_dhJj · 2024-11-02

**Soundness:** 2
**Presentation:** 3
**Contribution:** 2
**Rating:** 5
**Confidence:** 3

**Summary:**

This paper introduces GEOMATH, a multimodal mathematical reasoning benchmark specifically designed for the remote sensing (RS) domain, addressing the gap in mathematical reasoning capabilities of existing vision-language models (VLMs) in RS tasks. The main contributions are as follows:
Benchmark Development: GEOMATH consists of 3,773 high-quality vehicle-related questions spanning six mathematical subjects and 20 topics, all collected by drones from various altitudes and perspectives. Despite limited geographical coverage, the problems are rigorous and diverse due to comprehensive access to RS image parameters and detailed vehicle information.
VLMs Performance Evaluation: The paper conducts a comprehensive quantitative evaluation of 14 prominent VLMs, highlighting that solving these math problems requires high-resolution visual perception and domain-specific mathematical knowledge, presenting a challenge even for state-of-the-art VLMs.
Error Analysis: The authors explore the impact of image resolution and zero-shot prompting strategies on model performance, analyzing the reasons behind reasoning errors in GPT-4o.
Knowledge Transfer Capability: By comparing InternVL2 and GPT-4o, the study finds that the latter exhibits some degree of cross-view knowledge transfer capability.

**Strengths:**

1. The paper introduces the GEOMATH benchmark, specifically designed to evaluate the mathematical reasoning capabilities of vision-language models (VLMs) in remote sensing (RS) imagery, filling a gap in the existing research in this field.
2. It conducts a systematic evaluation of 14 prominent models, providing an in-depth comparison of their performance in mathematical reasoning tasks.

**Weaknesses:**

Despite the GEOMATH benchmark providing an effective evaluation tool for mathematical reasoning in the remote sensing field and laying a foundation for future research, there are still several issues:

1.The paper lacks innovation, as it does not propose effective methods for fully leveraging the capabilities of the GEOMATH benchmark in the remote sensing domain.

2.While the paper analyzes the strengths and weaknesses of existing methods on the GEOMATH benchmark, it does not specify how these methods could be improved.

3.There is limited comparative analysis of the GEOMATH benchmark with other datasets, and the paper does not provide detailed related research discussions.

**Questions:**

The paper analyzes the strengths and weaknesses of existing methods on the proposed GEOMATH benchmark, such as the limitations of advanced models like GPT-4o when lacking domain-specific mathematical knowledge. However, it does not indicate how to improve these methods or explain the reasons behind these issues.
There is limited comparative analysis of the GEOMATH benchmark with other datasets; presenting a comparison of key statistics in tabular form could enhance clarity regarding the differences among datasets.
The discussion of related work in the paper is relatively sparse, which could be expanded to provide a more comprehensive context.

---

### Official Review · Reviewer_21gH · 2024-11-04

**Soundness:** 3
**Presentation:** 3
**Contribution:** 3
**Rating:** 6
**Confidence:** 3

**Summary:**

The paper introduces GeoMath, a multimodal mathematical reasoning dataset specifically designed for the remote sensing (RS) domain, addressing a significant gap in existing datasets that primarily focus on visual perception without incorporating domain-specific mathematical knowledge. Although GeoMath is limited in size, it encompasses a wide range of complex mathematical concepts such as geometry, algebra, and arithmetic, which remain largely unexplored in other datasets. The study includes a comprehensive evaluation of 14 vision-language models (VLMs) and various zero-shot prompting strategies, revealing that even the most advanced models show notable limitations in RS-specific mathematical reasoning. The paper provides valuable insights into the performance of these models, including findings that higher image resolution does not yield as substantial a performance boost as expected, and that accuracy decreases as the number of reasoning steps increases. With the highest accuracy achieved not exceeding 35%, the results highlight the need for developing VLMs capable of handling domain-specific, mathematical, and multimodal reasoning tasks in RS applications.

**Strengths:**

1. Introduces a dataset for complex mathematical concepts requiring domain-specific knowledge, not comprehensively present in previous Remote Sensing VQA datasets.
2. Comprehensive dataset: The benchmark covers 11 distinct 4K resolution RS scenes, with varying combinations of drone’s above ground level (AGL) and pitch angles. It covers 6 mathematical subjects and 20 topics.
Contains reasoning steps to answer each question, ranging from a minimum of 2 to a maximum of 6. First dataset to do so in the RS domain. Includes metadata such as camera parameters and vehicle information, enabling other downstream tasks. Includes 4K high-resolution images. Versatile: covers various remote sensing applications such as surveying, surveillance, and market research
3. Offers an in-depth tabular and visual comparison with existing RS datasets, emphasising the need and importance of this work.
4. The authors provide analysis on the impact of image resolution and the number of reasoning steps on the VLM model’s performance
5. Implements zero-shot prompting techniques and qualitative comparisons between various models and their performance

**Weaknesses:**

1. The dataset is limited in size, with only 360 images and 424 unique questions, which may raise concerns given the goal of evaluating VLMs. While the extensive details and high quality of the dataset make expansion challenging, the small dataset size could impact its generalisability.
2. The scope of subjects and application scenarios considered is limited. Although this is understandable within the current scope of the work, expanding this aspect in future iterations would be beneficial.
3. The authors do not provide clear details on the evaluation methodology for free-form questions, which comprise 57.8% of the dataset. Clarification on how string or list outputs are managed and how accuracy is calculated for these questions would strengthen the paper.
4. Zero-shot evaluations, while helpful for initial insights, are less informative due to the inherent limitations of VLMs in handling out-of-distribution (OOD) tasks. VLMs often perform well on familiar training images but struggle with OOD data. More results on models fine-tuned on related RS datasets would provide a more comprehensive analysis. Although the authors include one VLM fine-tuned on RS data, extending this to other VLMs with fine-tuning on previously published RS datasets would enhance the comparison and yield richer insights.
5. The dataset’s reliance on specific vehicle attributes and limited scene diversity may restrict its applicability to other RS tasks beyond those involving vehicles. This could be addressed in future iterations by including broader scenarios or diverse object types.

Some minor comments:
1. In Figure 2, there are no visible vehicles in the image. While the figure effectively demonstrates the problem, adding visible examples would enhance its intuitiveness and improve reader understanding.
2. Detailed information about data collection could be moved to the appendix, allowing the main paper to emphasise the analysis of the various VLM models and their performance.
3. In Section 3.2, the paper mentions having three experimental setups, but only two are detailed.

**Questions:**

I would like some more information regrading the evaluation of the free form questions. Rest all looks good.

---

### Note · Authors · 2024-11-13

I have read and agree with the venue's withdrawal policy on behalf of myself and my co-authors.